# Classification of Cross-Country Ski Skating Sub-Technique Can Be Automated Using Carrier-Phase Differential GNSS Measurements of the Head’s Position

**DOI:** 10.3390/s21082705

**Published:** 2021-04-12

**Authors:** Øyvind Gløersen, Matthias Gilgien

**Affiliations:** 1Department of Physical Performance, Norwegian School of Sport Sciences, 4014 Oslo, Norway; matthiasg@nih.no; 2Center of Alpine Sports Biomechanics, Engadin Health and Innovation Foundation, 7503 Samedan, Switzerland

**Keywords:** XC-skiing, neural network, machine learning, artificial intelligence, GPS, technique, outdoor, performance, sport

## Abstract

Position–time tracking of athletes during a race can provide useful information about tactics and performance. However, carrier-phase differential global navigation satellite system (dGNSS)-based tracking, which is accurate to about 5 cm, might also allow for the extraction of variables reflecting an athlete’s technique. Such variables include cycle length, cycle frequency, and choice of sub-technique. The aim of this study was to develop a dGNSS-based method for automated determination of sub-technique and cycle characteristics in cross-country ski skating. Sub-technique classification was achieved using a combination of hard decision rules and a neural network classifier (NNC) on position measurements from a head-mounted dGNSS antenna. The NNC was trained to classify the three main sub-techniques (G2–G4) using optical marker motion data of the head trajectory of six subjects during treadmill skiing. Hard decision rules, based on the head’s sideways and vertical movement, were used to identify phases of turning, tucked position and G5 (skating without poles). Cycle length and duration were derived from the components of the head velocity vector. The classifier’s performance was evaluated on two subjects during an in-field roller skiing test race by comparison with manual classification from video recordings. Classification accuracy was 92–97% for G2–G4, 32% for G5, 75% for turning, and 88% for tucked position. Cycle duration and cycle length had a root mean square (RMS) deviation of 2–3%, which was reduced to <1% when cycle duration and length were averaged over five cycles. In conclusion, accurate dGNSS measurements of the head’s trajectory during cross-country skiing contain sufficient information to classify the three main skating sub-techniques and characterize cycle length and duration.

## 1. Introduction

Global Navigation Satellite Systems (GNSS) are frequently used to study performance in racing and endurance sports. The results can be presented live to spectators or used post session in performance or tactical analyses for practitioners and scientists. A performance analysis typically entails a continuous comparison of time gained or lost between competitors throughout the race [1]. However, knowledge about when and where athletes gain or lose time compared to their opponents has limited value unless the reasons for the time loss or gain can be elucidated. In cross-country ski racing, factors explaining differences in performance include physiological, tactical (i.e., pacing [2]) and technical aspects. Recent studies have shown that estimating metabolic energy expenditure [3,4] and conducting performance analyses [1] can be achieved by using GNSS measurements.

Basic variables reflecting an athlete’s technique, such as cycle duration, cycle length and choice of sub-technique, can also be measured by wearable sensing technology. In the scientific literature, this has most frequently been attempted using inertial measurements units (IMUs). An IMU-based analysis system capable of detecting cycle length and duration has been developed for some sub-techniques in classical skiing [5]. However, this approach is challenging to transfer to ski skating techniques, since skating, in contrast to most classical skiing sub-techniques, lacks stationary phases in the motion pattern. Stationary phases help facilitate drift corrections in the IMU displacement calculations. Although substantial technological advancements have been made within both microelectromechanical sensor (MEMS) design and attitude drift-correction algorithms [6,7], we are not aware of a combination of consumer-grade MEMS-sensors and drift-correction algorithms capable of providing drift-free displacement measurements during a sliding gate, such as ski skating. For instance, conventional techniques used for human gait, such as pedestrian dead reckoning algorithms, would be difficult to implement for a sliding gait. Therefore, technological solutions to measure cycle length in ski skating using IMUs would most likely require fusion with complementary measurements, e.g., GNSS position measurements. Manual and automated sub-technique classification has also been attempted using IMUs, both in classical [8,9,10,11,12] and skating styles [11,12,13]. However, studies attempting automated classification from IMU measurements conclude that a relatively large number of sensors attached to different landmarks on the athlete’s body are necessary to achieve reasonable classification accuracy. Specifically, 74–97% accuracy was obtained using five IMU sensors and 65–79% accuracy was achieved using one sensor [11]. Hence, IMU-based calculation of cycle length, cycle duration and sub-technique classification for ski skating has two challenges: (i) position sensors are required to correct for drift, and (ii) multiple IMUs mounted on different locations are required, making it cumbersome to use in training and competition applications. It is therefore worthwhile investigating whether sub-technique, cycle length, and duration can be derived from kinematic carrier phase differential GNSS (dGNSS) measurements directly, as an alternative to IMU-based methods. The use of kinematic dGNSS might be a reasonable alternative, since the technology can provide sub-decimeter position accuracy and high temporal resolution [14]. Using dGNSS to obtain drift-free measurements of step length has already been proposed for conventional gaits, such as walking [15]. If the parameters required to describe skating techniques could be extracted from dGNSS position measurements, dGNSS could be applied as a one-sensor system capable of generating the data needed for performance analysis (time loss or gain) [1] and its underlying technical, tactical and physiological [3,16] explanatory variables.

The idea of using kinematic dGNSS measurements to classify skiing sub-technique has previously been explored in classical cross-country skiing [17]. The authors of that study manually classified different sub-techniques by visual inspection of the position displacement curves. The fact that they achieved a high classification accuracy indicates that dGNSS position measurements contain sufficient information for classifying the sub-technique that was used, at least for classical skiing. However, this has not yet been investigated for ski skating styles. Furthermore, an automated rather than manual method is required to increase the applicability of the approach in practice. Therefore, the aim of this study was to develop an automated method to extract cycle length, duration, and sub-technique from kinematic dGNSS position measurements during cross-country ski skating, and to validate the method’s performance in a race-like situation.

## 2. Materials and Methods

The design of this study was composed of two parts: a method development using measurements from indoor treadmill roller skiing, and an adoption and validation of the method using dGNSS measurements from a roller skiing test race in a race-like outdoor course. The measurements used in both parts are from previous studies conducted by our group [4,18,19]. These studies were approved by the ethics committee at the Norwegian School of Sport Sciences (ref. 02-020517) and the Norwegian Centre for Research Data (ref. 54257), and were conducted in accordance with the Declaration of Helsinki and Norwegian law. Data analysis was conducted in Matlab R2019a (The Mathworks Inc., Natick, MA, USA), except for processing of dGNSS measurements and 3D optical motion capture measurements. These were conducted using dedicated software, as described in the text.

### 2.1. Method Development Using Optical Motion Capture Data during Indoor Treadmill Roller Skiing

The participants for the indoor treadmill roller skiing-based method development were 6 elite male skiers (age 26 ± 2 years, height 181 ± 5 cm, weight 79.5 ± 5 kg). During the indoor treadmill skiing, the athletes’ movement patterns were captured at 250 Hz using a 9-camera setup (Oqus 400, Qualisys AB, Gothenburg, Sweden) and 41 retroreflective markers distributed on each participant’s skin and equipment. Calculation of marker positions from the camera system’s measurements was conducted in Qualisys Track Manager (Qualisys AB, Gothenburg, Sweden), and included a dynamic calibration with resulting accuracy of <2 mm RMS. The participants completed trials under two different conditions for each of the 3 main sub-techniques, termed G2, G3 and G4 following the convention in Nilsson et al. [20]. The three main sub-techniques, along with the less frequently used G5, are illustrated in Figure 1. The conditions were 4° and 6° at 3 m/s for G2 and G3, and 4 m/s and 5 m/s at 3° for G4. Fifteen movement cycles were recorded for each condition.

To generate a signal resembling the expected dGNSS position measurements of the head trajectory, we first transformed the marker position measurements into a global coordinate frame (GCS) that was stationary with respect to the treadmill band. This was done by numerical integration of the treadmill’s instantaneous velocity. The treadmill velocity was controlled by a feedback loop with an accuracy of 0.04 ± 0.01 m·s^−1^ in the 1–10 m·s^−1^ speed range. Second, we decimated the optical motion capture measurements of a marker placed on the head (the superior part of the neurocranium, Figure 2A) to 50 Hz (by keeping every 5th frame) and added pseudorandom white noise with a standard deviation of 30 mm (Figure 2B). The resulting signal, termed the dGNSS proxy, had similar time resolution and spatial accuracy to what can be expected from multifrequency, carrier-phase dGNSS measurements operating in kinematic mode [14]. The dGNSS proxy was then fitted with a smoothing spline filter commonly applied to dGNSS measurements [21]. This smoothing spline was used in the further analyses to detect cycle duration, cycle length, and classify the sub-technique used.

The head’s sideways oscillations were used to define the cycle’s start and end points. This was done because all ski skating techniques (except the tucked position) involve one complete sideways oscillation of the body per technique cycle. To allow generalization to field conditions with frequent changes in skiing direction, we defined “sideways” to be perpendicular to both the skiing direction and vertical direction (gravity). Specifically, we defined a local coordinate system (LCS) whose origin was the head’s trajectory after being passed through a 0.3 Hz low pass filter (5. order bi-directional Butterworth). The 0.3 Hz low pass filter applied to the head trajectory removed frequencies attributed to the cyclic movement patterns used for propulsion (whose dominant frequencies typically are between 0.5 and 1 Hz), while allowing slower frequencies attributed to the progression through a ski course to pass through [1]. Skiing direction was defined as parallel to the LCS velocity vector (Figure 3) Peak sideways velocity of the head (expressed in the LCS) was used to define the cycle’s start and end points (Figure 2C). Only sideways velocity peaks with prominence of ≥0.7 m/s and temporal separation between peaks of ≥0.8 s were accepted as start and end points of cycles. Cycle duration was defined as the time between two succeeding peaks, and cycle distance was defined as the Euclidean distance between the head’s position in the GCS (spline-filtered dGNSS proxy) at the start and end of each cycle (Figure 2D). The cycle duration and distance calculated from the head’s trajectory were validated against time between succeeding pole plants and the Euclidean distance between the body’s center of mass (CoM) at two succeeding pole plants. A pole plant was defined as peak acceleration of a marker on the left pole and CoM was calculated using a 19-segment rigid body model, as described in Myklebust et al. [19].

To obtain waveforms that were characteristic for each sub-technique yet relatively unaffected by speed or anthropometric differences, we first resampled the head’s trajectory during each cycle to 12 time points (Figure 4, top row). We then cross-correlated both the vertical and the fore/aft components of the head’s displacement with mediolateral displacement, using a maximal lag of half a cycle (6 points). This resulted in two 13-point waveforms that clearly distinguished the 5 ski skating techniques, including left/right symmetric variants of G2 and G4 (Figure 4, bottom row). These waveforms were concatenated to a 26-dimensional feature vector and used to train a feed-forward neural network classifier (NNC). A feed forward NNC is an appropriate classifier for this problem, because the features are not independent of each other. The fully connected neurons of an NNC are capable of learning these interactions. Furthermore, time normalization to a fixed length feature vector characterizing each technique cycle simplifies the classification problem, omitting the need for a recurrent NNC (which would be more sensitive to overfitting and therefore require more training data). The NNC was set up with one hidden layer of 15 neurons and 5 output classes, corresponding to the 5 sub-techniques. It was trained using Bayesian regularization backpropagation on 510 recorded technique cycles, of which 45 were G2R (G2 right side), 120 G2L (G2 left side), and 165 G3, 120 G4R and 60 G4L from the 6 different participants. To improve generalizability, a random subset of 20% of the cycles was used as validation data.

### 2.2. Validation Using Kinematic dGNSS Measurements during an Outdoor Roller Skiing Test Race

The data used in the indoor method development represented an ideal situation for classification, because treadmill roller skiing at constant velocity and incline introduces little cycle-to-cycle variability and few cycles deviating from the sub-technique characteristics due to changes in terrain incline or course direction. To validate the method’s performance in an ecologically relevant field situation, we therefore tested the NNC on real dGNSS measurements from two skiers during a 13.5 km test race on an outdoor roller skiing course. The participants were equipped with a dGNSS consisting of an antenna mounted on the skier’s helmet (G5Ant-2AT1, Antcom, Torrence, CA, USA) connected to a dual frequency (L1 + L2) GPS/GLONASS receiver (Alpha-G3T, Javad, San Jose, CA, USA) placed in a small backpack (Figure 3). The total weight of the dGNSS system was 940 g (receiver 430 g, backpack 350 g, antenna 160 g). A stationary base station (GrAnt-G3T antenna and Alpha-G3T receiver, Javad, San Jose, CA, USA) was placed close to the course to facilitate differential positioning. The position solutions were calculated using kinematic carrier phase double difference solutions at 50 Hz using geodetic post-processing software (Justin, Javad GNSS Inc., San Jose, CA, USA), and filtered using the same smoothing spline filter [21] as described in the method development paragraph. The positioning accuracy of this combination of antennae, receivers and processing procedure has been reported to be about 5 cm for fixed ambiguity solutions in comparable GNSS conditions [14].

The dGNSS position measurements were analyzed using the same filtering, cycle detection and NNC algorithms outlined in the method development paragraph. However, some additional steps were added to detect techniques used during ski skating outdoors that were not performed, and could therefore not be detected, in the standardized indoor treadmill trials. Specifically, this included techniques used for turning (termed “Turn”) or downhills. The latter included both the tucked position to minimize air drag (termed “Tuck”) and free skating, where athletes only used the legs to generate propulsion (termed “G5”, Figure 1). In “Tuck”, skiers assumed an aerodynamic position by squatting, tilting the upper body forward and keeping the poles along the body. Hence, it is not comprised of cyclic mediolateral oscillations, and the cycle detection algorithm based on sideways velocity cannot be applied to detect periods of Tuck. In contrast, G5 and Turn do involve mediolateral oscillations of the head, and therefore can be submitted to the cycle detection algorithm. However, as both techniques are distinct from the three main gears (G2–G4), they should not be classified by the NNC, which was trained for these gears only. Including Turn and G5 in the NNC treadmill training data was not attempted due to the following considerations: Turning can be accomplished in several ways, i.e., with and without use of poles. The lack of a consistent movement pattern for all turning techniques means that Turn would be difficult to learn for the NNC. G5 is more consistent but is infrequently used. Including G5 in the NNC classifier along with the more prevalent G2–G4 sub-techniques might reduce overall classification performance (by misclassifying other sub-techniques as G5).

Hence, rather than expanding the NNC with additional classes (that are either infrequent or not cyclic), we added the following analytical steps to detect these movement patterns prior to NNC classification. The respective analytics are shown graphically in the decision tree in Figure 5. First, to determine if the athlete was in the Tuck technique, we performed a Short-time Fourier transform (STFT) on the head’s velocity expressed in the LCS. The STFT used a 256 sample Hanning window and 512 sample FFT length, with 255 sample overlap. This window length corresponded to 5.12 s, which provides adequate frequency resolution while including relatively few technique cycles within one window (typically 2–4 cycles, depending on speed and sub-technique). The tradeoff between frequency resolution and temporal resolution is further addressed in the discussion. The 255 sample overlap ensures that the STFT is performed once for each GNSS position measurement. The other extreme would be zero overlap, in which case the STFT would be performed with non-overlapping time windows, i.e., every 256th GNSS position measurement. Computational cost could be reduced with a smaller overlap and by interpolating between the decimated time points; however, this was not investigated in the current study. The magnitudes of frequencies between 0.5 and 1.0 Hz in the sideways oscillations were summed, and if the sum was less than a fixed threshold of 40 m/s, the sub-technique was defined as Tuck. This threshold was based on the distribution of the summed frequency spectrum, which had a minimum at approximately 50 m/s. Second, all measurements not classified as Tuck were sent to the cycle detection algorithm for classification as Turn or G2–G5. Sub-technique cycles were defined as Turn if the rate of change in skiing direction during a cycle exceeded a threshold *T*_turn_, which was set to 10°/s by default. Third, to detect G5, the STFT of the head’s vertical velocity component was used to detect whether there was substantial use of the upper body. If the summed magnitude of frequencies between 0.5 and 1.5 Hz in the vertical oscillations was less than a threshold of *T*_G5_ = 100 m/s, these cycles were defined as G5. Finally, cycles that were not classified as Turn or G5 were sent to the NNC and classified as G2–G4 using the NNC procedure, as explained in the indoor development section. Because the value of the chosen thresholds affects the classification results, the classifier’s sensitivity to changes in *T*_G5_ and *T*_turn_ was assessed by changing both variables systematically (from 70 to 130 m/s and 7 to 13°/s for *T*_G5_ and *T*_turn_, respectively), and calculating the fraction of cycles that were correctly classified for each combination of parameters. The threshold for detecting sideways oscillations was not included in the sensitivity analysis, because this would then change the number of detected cycles. This is a challenge because the manual validation (described in the next paragraph) would need to be repeated for every threshold tested, which is not feasible from a practical viewpoint.

The automated classification algorithm was validated by comparison with manual classification based on video recordings from a small video camera (Hero 3, GoPro, San Mateo, CA, USA) taped to the participant’s chest. We created a subtitle file for each participant that was overlaid on the video images and displayed technique and classification information using VLC 3.0.11 (VideoLAN, 18 rue Charcot, 75013 Paris, France). Specifically, the subtitle was updated for each detected sub-technique cycle, and displayed the classified technique, accumulated cycle count, and GNSS ambiguity resolution status (fixed or float). Classification performance was evaluated by matching all technique cycles recorded by the manual and automated classifications. This was performed by manually adding missed cycles to a “none” category for the automated classifications and adding falsely recorded cycles to a “none” category for the manually detected cycles. The results are presented as confusion matrices. Time periods where fixed ambiguity solutions could not be established were omitted from the results. This was done because position measurement accuracy in these situations was severely deteriorated [14], such that sub-technique classification using the head’s displacement was not feasible. Fixed ambiguity solutions were found for 90.0% of all dGNSS measurements; hence, 10.0% of the measurements were discarded and cycles were not classified for these periods.

## 3. Results

### 3.1. Method Development Using Optical Motion Capture Data during Indoor Treadmill Roller Skiing

The relative root mean squared (RMS) deviations between cycle duration calculated from the head’s trajectory and pole plants when applied to individual cycles on the treadmill data were 2.1 ± 0.3%, 2.1 ± 0.6% and 3.0 ± 1.1% for G2, G3 and G4, respectively. However, RMS deviation was reduced when averaged over N succeeding cycles, and the rate of reduction exceeded the N^−1/2^ behavior expected from independent (i.e., not successive) samples (Figure 6, top row). Specifically, the RMS deviation was ≤1% when averaged over five successive cycles for all three sub-techniques. The results were similar for cycle length (1.9 ± 0.3%, 2.2 ± 0.7% and 3.1 ± 1.0% for G2, G3 and G4, respectively), indicating that the discrepancies were mostly due to the estimates of cycle duration and not due to differences in the displacement of the head and CoM (Figure 6, bottom row).

After training, the NCC correctly classified all cycles from the treadmill measurements, both those used as training data (408 cycles) and those used as validation data (102 cycles).

### 3.2. Validation Using Kinematic dGNSS Measurements during an Outdoor Roller Skiing Test Race

Confusion matrices showing the automated classification algorithm’s performance compared to manual classification are shown in Figure 7. For the three main sub-techniques classified by the NCC, classification sensitivity (i.e., fraction of cycles correctly classified) was between 92.1% and 97.1%. Classification performance for the movement patterns that were not classified by the NNC (G5, Turn and Tuck) was comparatively poorer, ranging from 32.0–88.1%. Of these, classification of Tuck was most successful (88.1% sensitivity). Furthermore, the distribution of mediolateral oscillation magnitude (Figure 8A) used as a criterion to classify Tuck clearly showed a distinct minimum close to the decision threshold. The peaks on both sides of the minimum presumably correspond to either Tuck (very little sideways oscillations) or all other skating styles (substantial sideways oscillations). Turn had a classification sensitivity of 74.9%, and the distribution of change in skiing direction (Figure 8C) did not show any obvious threshold region. Lastly, G5 was frequently misclassified as either G2, G3 or G4, indicating that step 3 in the classification tree (Figure 5) was unsuccessful in reliably separating G5 from the main sub-techniques passed to the NNC. Although the distribution of vertical oscillation (Figure 8B) showed a distinct minimum, presumably separating periods of Tuck and G5 (both showing very little vertical oscillation) and other skating styles, systematic variation of the threshold within this region was unable to notably increase the number of successful classifications (Figure 8D).

## 4. Discussion

This study shows that dGNSS measurements of the head’s trajectory contain sufficient information to classify the three main sub-techniques (G2–G4) used in cross-country ski racing, and that the classification can be automated using a feed-forward NNC. Measurements of cycle length and duration can also be extracted from the position measurement, and show good agreement with gold standard methods, at least if averaged over five cycles or more. The NNC rarely (≤5% of the cycles) confused any of the three main gears, even in a race-like situation with frequent changes in sub-technique, skiing direction and incline. Hence, this study extends the idea presented by [17] to ski skating techniques and proposes an automated method for sub-technique classification and measurement of cycle length.

Classification of Tuck, Turn and G5, which was achieved using hard decision rules, was not as good as the NNC-based classification of the three main gears (G2–G4). G5 in particular was often misclassified, and the most common cause was that cycles which should have been classified as G5 were passed on to the NNC. This implies that vertical oscillations of the head during G5 frequently exceeded the threshold *T*_G5_. However, raising this threshold would result in several misclassifications of the three main gears as G5, leading to a reduced overall performance of the classifier (Figure 8D). It is possible that classification performance could be improved by reducing the window length of the STFT, since the 5.12 s window length makes it difficult to detect transitions from G5 to a lower gear or vice versa. However, due to the Gabor limit [22], a shorter window would result in a lower frequency resolution, which could hinder identification of the frequencies associated with typical cyclic movements of the torso (0.5–1.5 Hz). Wavelet transforms have a better time-frequency resolution than the STFT and might therefore lead to better classification performance. Another possibility to improve G5 classification would be to include G5 in the training data of the NNC, rather than to classify it from the hard decision rules in this study. However, this would increase the NNC’s complexity, which increases the probability of misclassifications (keeping the training data constant). Furthermore, even if the NNC would have a higher sensitivity for classifying G5 compared to the simple threshold method used in the current study, overall classification performance (defined as the number of correctly classified cycles) might deteriorate because the more frequently used G2–G4 can be misclassified as G5.

The NNC was trained on measurements from six different skiers and subsequently tested on two other skiers in different conditions (in-field vs. treadmill skiing). The NNC’s high classification accuracy in diverse ecological conditions, using a different sensing technology (dGNSS vs. optical motion capture) and being employed on skiers it had not been trained on, indicates that the NNC is generalizable, at least to skiers with similar skill levels. “Similar skill level” in this context refers to competitive skiers on a high national level (the training data) and recreational skiers with background as competitive skiers (the in-field validation data). It might be that the NNC’s performance will degrade if the method is applied to skiers without a background as competitive skiers, who are likely to exhibit a larger variability in their movement patterns. In contrast to the NNC, the thresholds used in the hard decision rules were determined based on data from the two participants skiing in-field. Therefore, their generalizability to other skiers is unclear. The sensitivity analysis presented in Figure 8D shows a reasonably wide plateau with stable classifications for *T*_G5_ and *T*_Turn_. Specifically, both thresholds could be varied within a range of 30–50% of their optimal values without overall classification performance falling more than 1 percentage point. For the threshold used to detect Tuck, there was a distinct valley from 35–65 m∙s^−1^ in the sideways oscillation magnitude. This valley seemed consistent for both skiers (Figure 8A), and the threshold used in the current study (40 m∙s^−1^) was within this valley. Taken together, this shows that classification performance was not very sensitive to modest changes in the threshold values for the skiers included in the current study. Nonetheless, the proposed method’s generalizability, particularly with respect to the hard decision rules and different skill levels of skiers, cannot be rigorously established from the data in the current study. In particular, the initial threshold values were set by visual inspection of the histograms in Figure 8A–C. Since these histograms are based on the data of the two participants in the current study, there is a risk of sampling artefacts that do not generalize to a broader population.

Another point that might have affected generalizability is the weight of the GNSS receiver (780 g, including backpack) or antenna (160 g, excluding helmet), which might have caused some alterations of the movement patterns. However, as none of the athletes complained about the system affecting the execution of their test race, even though they were skiing at racing speed, we assume that the effect of this additional mass was small.

Generalizability from roller skiing to skiing on snow cannot be decisively established from the data in the current study. Unfortunately, direct comparisons between skiing on snow and roller skiing are lacking in the literature; however, the dissertation of Myklebust [23] includes some data on this. He reported some differences in the kinematics of the pelvis (rotational and mediolateral amplitudes) in the G3 technique when comparing skiing on snow to roller skiing, at equal speeds and inclines. However, these changes were small, and in magnitude comparable to changes associated with different rolling resistances during roller skiing. Considering that the classification algorithm proposed in the current study performs well for both treadmill and in-field roller skiing, and over a large range of speeds and inclines, we find it likely that the classification algorithm should perform similarly for skiing on snow as for roller skiing.

Most previous studies on technique classification in cross-country skiing have used IMUs to classify sub-technique selection [8,9,10,11,12,13,24,25,26,27]. Of these studies, two have attempted automated classification of ski skating techniques in ecological (i.e., in-field) conditions comparable to the current study [11,13]. Sakurai et al. [13] tested an algorithm for automated classification during roller skiing at racing speeds in a roller skiing course. The algorithm was based on hard decision rules and used data from four IMUs (accelerometers and gyroscopes) mounted on the skis and wrists. They reported an accuracy of 87–98% for the three main gears (G2–G4). This performance is comparable to the current study, perhaps except for the classification of G2, which was higher in the current study (97% vs. 87%). However, Sakurai et al. [13] evaluated their algorithm on both sexes and more subjects (*N* = 15) than the current study and, therefore, the results are more likely to be generalizable to a larger group. More recently, Jang et al. [11] tested several different IMU sensor configurations, with the number of sensors ranging from 1 (pelvis only) to 17. They tested four different skiers at a freely chosen pace in a competition skiing course, using a leave-one-out approach to independently train and test their classifier (which was a Long Short-Term Memory Convolutional Neural Network). They concluded that a setup with 5 IMUs (pelvis, both hands and feet) was the optimal configuration; additional sensors did not increase performance substantially, but a single sensor mounted on the pelvis was insufficient for obtaining acceptable results (on average 44% accuracy). Classification accuracy was, on average, 91% for the 5-sensor setup, which is similar to the results using a single but accurate position measurement sensor, as shown in the current study. While their classifier included both classical and skating styles (eight classes in total—four classical and four skating), making the classification problem challenging, they also manually removed transition, turning, and downhill phases from the data, making the classification problem easier. In summary, although direct comparisons between studies is difficult due to methodological differences, it seems that IMU-based methods with ≥4 sensors distributed on the athlete’s body provide classification performance comparable to the current dGNSS based approach. Acceptable classification accuracy using <4 IMUs has not been demonstrated in ecological situations; however, two studies have reported promising results with more minimalistic IMU sensor setups (one or two sensors) during treadmill skiing [24,27]. Roller skiing in a competition course introduces substantially more variability in the movement pattern than what is present in treadmill roller skiing. For instance, the NNC classifier in the current study successfully classified all cycles during treadmill skiing, but accuracy was reduced to 92–97% in ecological conditions. Consequently, successful classification of ski skating techniques in ecologically relevant situations using only one or two IMUs has not yet been demonstrated.

Although IMU-based approaches for determining cycle length have been successfully applied to some classical style skiing techniques that include stationary phases [5], these approaches would not be transferable to ski skating techniques. In fact, obtaining cycle length from IMU measurements alone is non-trivial, even for conventional gaits with stationary phases [28]. Accurate position sensing technologies such as dGNSS have an intrinsic advantage over IMU-based calculations of cycle length, because displacement can be measured directly. This advantage is not limited to sliding gaits, and has also been suggested for conventional gaits [15].

Combined with earlier studies which have shown that carrier phase dGNSS measurements are capable of providing variables describing both performance [1] and instantaneous rate of energy expenditure [3,16], the results of the current study clearly show that dGNSS is an interesting technology for tracking ski-skating. The method presented in this study omits the requirements for mounting multiple sensors on the athlete [11,13] and for functional calibration of IMUs using constrained movements prior to measurement [7,29], which is time-consuming and, therefore, not very applicable in training and competition situations. This advantage comes at the expense of mounting a stationary GNSS unit close to the competition circuit for differential GNSS correction, and an increased form factor of the athlete-mounted sensor. The dGNSS antenna and receiver used in this study are larger and heavier than typical IMUs; however, newer models of dGNSS receivers have a substantially smaller form factor [1,17] and will be minimally disturbing to the athlete. Therefore, carrier phase dGNSS measurements can become a superior alternative to multi-sensor IMU systems for in-competition technique analyses.

A challenge with carrier-phase dGNSS measurements is that solution ambiguities must be fixed to obtain sub-decimeter kinematic position measurements. It is challenging to find fixed-ambiguity solutions throughout a race which often includes segments where GNSS signals are compromised, e.g., when athletes pass below bridges or through dense forest. In this study, which was conducted on a racetrack used in international competitions, ambiguities could not be fixed for 10% of the data. This shortcoming can be addressed by using GNSS receivers that can track signals from several GNSSs (the current study used signals from GPS and GLONASS, further systems include Beidou and Galileo), or through coupling with an inertial navigation system (INS) [21]. The latter would increase robustness against brief periods of poor GNSS signal quality and increase the fraction of the course where calculation of cycle characteristics and sub-technique classification is possible. In addition, an INS would return information on the angular movement of the body part it is attached to (the back). This would increase the information available for sub-technique classification, most likely also resulting in more accurate classifications in situations where GNSS signal quality is good.

In this experiment, the GNSS antenna was mounted on the head, which provided the best possible GNSS signal reception. If combined with an INS, positioning of the antenna on the upper back (e.g., in a pocket on the start bib) might be an acceptable solution because GNSS signal quality requirements are relaxed compared to position calculation without inertial corrections. Positioning on the upper back would be less intrusive for the athlete, but the movement characteristics specific to each sub-technique would change compared to the ones found in this study. Further studies are required to determine how ski-skating sub-techniques can be classified with alternative sensor locations, and by including angular information. The results from the current study suggest that a position sensor mounted on the upper part of the truncus (i.e., the neck) can contain sufficient information to classify most ski skating sub-techniques, and it is likely that classification performance can be improved further with angular information.

It is likely that classification approaches other than the one implemented in the current study would also have resulted in high classification accuracies. Therefore, we consider the main message of this study that information about sub-techniques, cycle length and cycle duration can be extracted from the dGNSS signal using an automated procedure, rather than to determine the optimal procedure to do so. However, it is our opinion that the method developed in this study is appropriate for the problem, and this is supported by a strong classification performance. A noteworthy feature is the cross correlation of the head’s trajectory with itself, which revealed waveforms that were highly distinct for each sub-technique, yet relatively independent of speed or cycle frequency. Feed-forward NNCs are appropriate classifiers for such feature vectors; however, other classifiers, such as support vector machines or random forest classifiers, might also have performed well. Recurrent neural networks offer the benefit of variable length feature vectors, omitting the need for time normalization. However, they would be at risk for overfitting without more training data, as the training data in the current study were relatively homogenous (different skiers, but very consistent conditions).

## 5. Conclusions

This study demonstrates that classification of ski skating sub-technique and determination of cycle length and frequency can be obtained from carrier phase dGNSS position measurements using an automated procedure based on feed-forward neural networks and hard decision rules. With this, dGNSS has become a method that can provide instantaneous information on performance [1], rate of energy expenditure [3] and technique, with a single sensor setup.

## Figures and Tables

**Figure 1 sensors-21-02705-f001:**
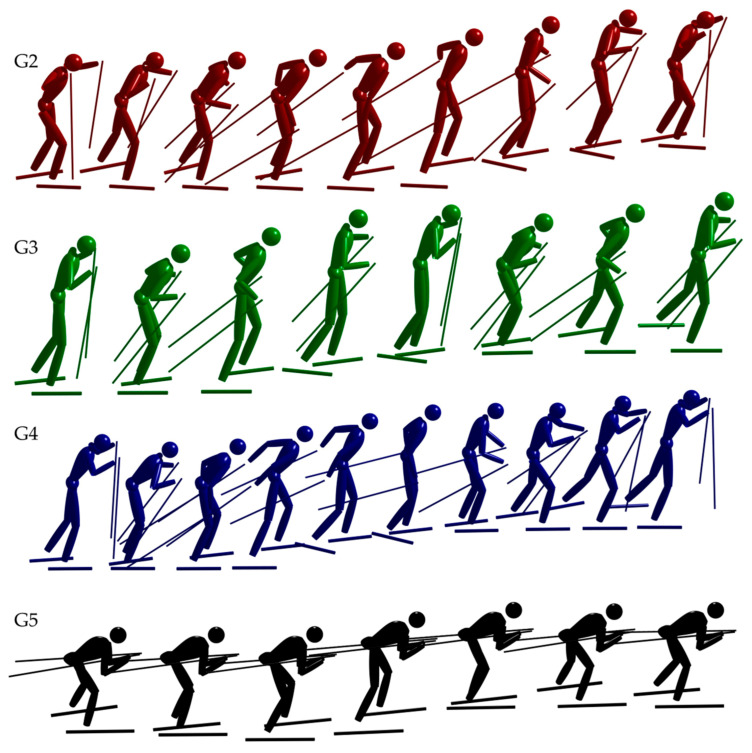
Illustration of one movement cycle of the ski-skating sub-techniques termed G2–G5. G2 and G4 consists of one poling cycle per movement cycle and can therefore be executed in left or right symmetric variants—this figure shows only the right symmetric variants. G3 and G5 have two or zero poling cycles per movement cycle, respectively, and do not have left/right symmetric variants. The color coding with red indicating G2, green indicating G3, and blue indicating G4 is used consistently throughout the manuscript.

**Figure 2 sensors-21-02705-f002:**
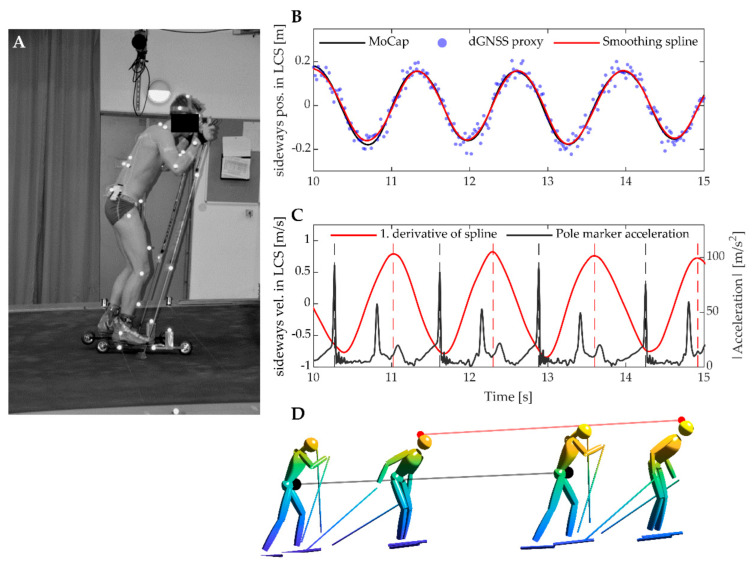
(**A**): Experimental setup used during the method development. A total of 41 retroreflective markers distributed on the participant’s skin and equipment were used to create a 19-segment rigid body model to estimate body center of mass (CoM) kinematics. The marker on the top of the head was used to generate a differential global navigation satellite system (dGNSS) proxy signal by downsampling to 50 Hz and adding pseudorandom white noise. (**B**): Sideways position of the dGNSS proxy (blue dots) and smoothing spline filter (red line), measured in the LCS (see manuscript body and Figure 3 for definition). (**C**): Peak sideways velocity of the dGNSS proxy signal was used to define cycle duration (dotted red lines), which was compared to cycle duration from pole ground contact (dotted black lines). (**D**): Cycle length (red line) was defined as the head’s displacement between two succeeding peaks in sideways velocity (i.e., dotted red lines in panel (**C**)). This was compared to the CoM’s displacement (black line) between two successive pole plants, or every second pole plant for the G3 sub-technique.

**Figure 3 sensors-21-02705-f003:**
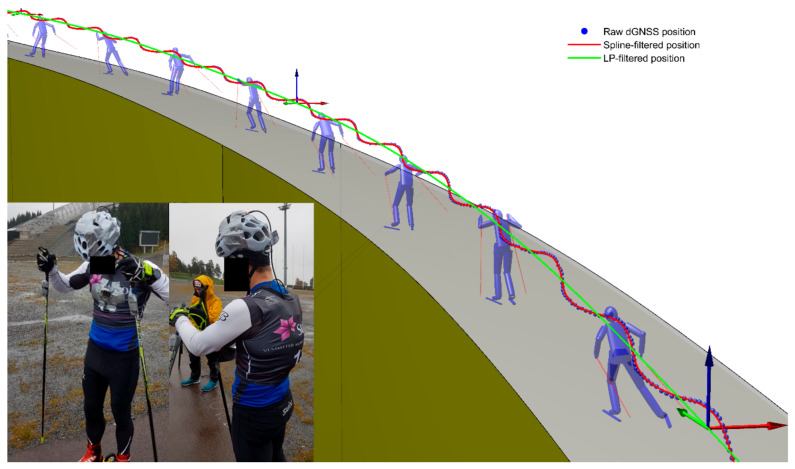
Definition of the local coordinate system (LCS) used to capture the head’s movement. Blue dots are raw dGNSS measurements, red line is the smoothing spline filter, and green line is the trajectory of the smoothing spline after being filtered with a 0.3 Hz low pass filter. The LCS origin followed the low-pass filtered trajectory (green line), and skiing direction was defined as parallel to this trajectory. Sideways was defined as the cross product of unit vectors along the skiing and vertical directions. Picture on the bottom left shows the experimental setup for the outdoor validation experiment (Section 2.2): the GNSS antenna was taped to the skier’s helmet while the receiver was carried in a small backpack under the start bib. A small camera was taped to the chest for validation of the sub-technique classification.

**Figure 4 sensors-21-02705-f004:**
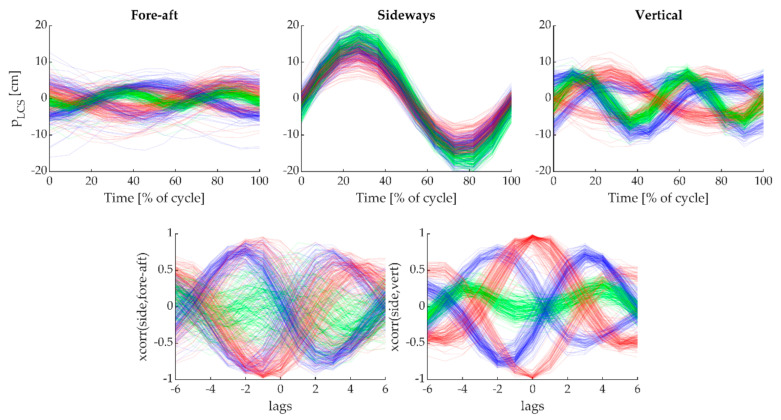
Data generated from the head position of the indoor treadmill trials. Top row: Head position normalized to a complete technique cycle (12 points per cycle) in the LCS. From left to right: fore-aft, sideways, and vertical directions. Colors: red, green and blue lines are G2, G3 and G4, respectively. Bottom row: waveforms generated by cross-correlating the curves in row 1. Left: fore-aft*sideways movement, right: vertical*sideways movement. The five main sub-techniques (including the left/right symmetric variants of G2 and G4) have distinct cross-correlation waveforms, particularly for the vertical*sideways component.

**Figure 5 sensors-21-02705-f005:**
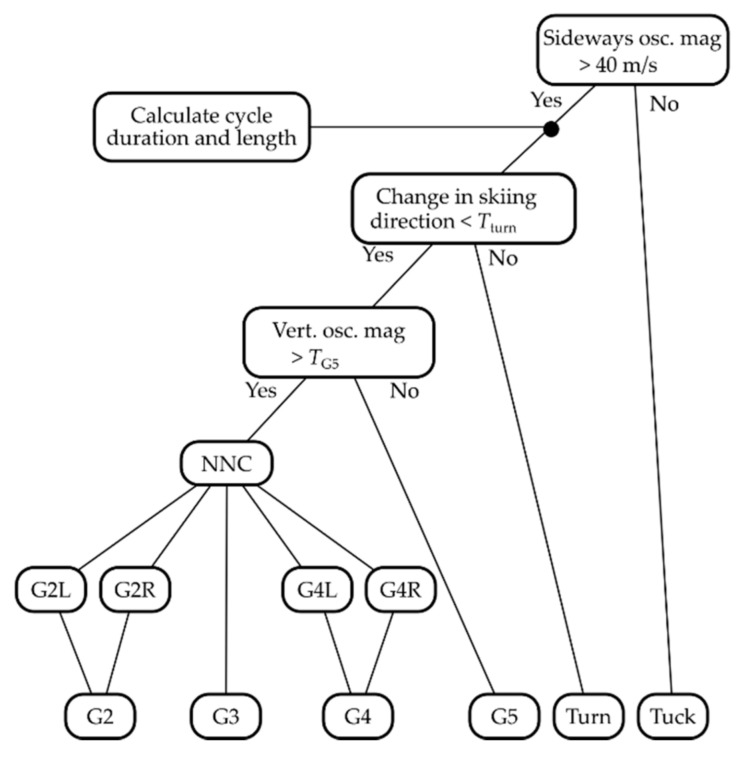
Classification tree for the field validation experiments. A Short-time Fourier transform STFT was used to check for oscillatory movement of the head and rate of turning prior to submitting the measurements to the NNC. Measurements were only passed to the NNC if oscillatory movement magnitude in both sideways and vertical directions was above given thresholds, and if the change in skiing direction was less than a threshold. The NNC classified the left/right symmetric variants of G2 and G4 individually, but these were merged to single classes for the validation, resulting in 6 sub-technique categories: G2, G3, G4, G5, Turn and Tuck.

**Figure 6 sensors-21-02705-f006:**
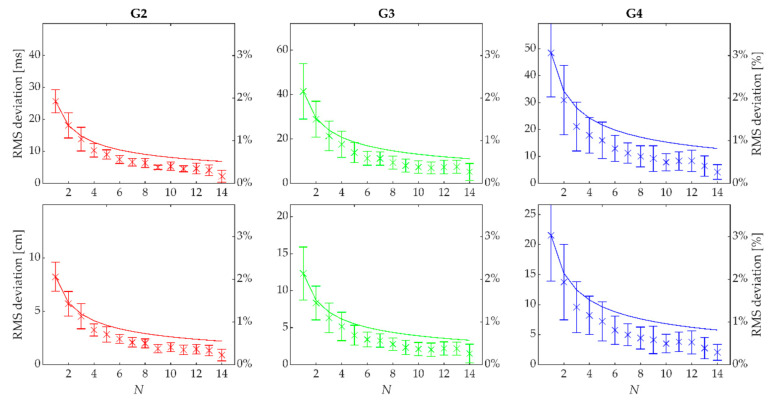
Root Mean Square (RMS) deviations between cycle duration (top row) and cycle length (bottom row) calculated using the head’s trajectory only and the criterion method (pole plants and CoM displacement), plotted as a function of the number *N* of successive cycles’ distance and time were averaged over (from 1 to 14 cycles). G2 is given in the left column, G3 in the middle column and G4 in the right column. The deviations are plotted in absolute values on the left *y*-axis (unequal scaling between time and distance plots) and relative values on the right *y*-axis (equal scaling between sub-techniques). The dotted lines indicate a function proportional to *N^−1/2^,* which would be expected when averaging over statistically independent cycles.

**Figure 7 sensors-21-02705-f007:**
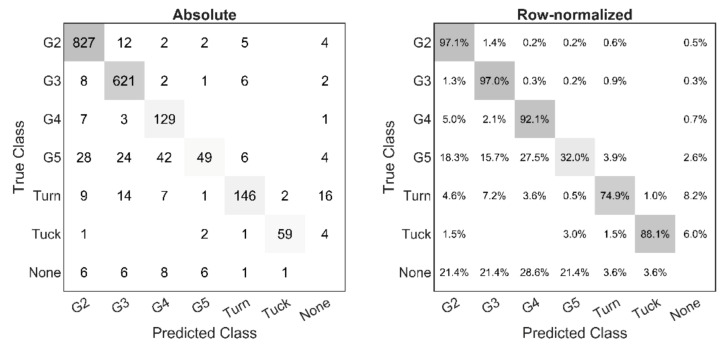
Left: Confusion matrix showing the automated classification algorithm’s performance during the roller skiing test race. Rows represent manual classification from video, columns represent the automated classification method. Diagonal entries show the number of correctly classified cycles while off-diagonal entries are instances where the predicted and true class differ. The rightmost column and bottom row represent cycles not detected by the automated algorithm and falsely detected cycles (i.e., not apparent in the video), respectively. Right: Same information as in left matrix but normalized to the number of true cycles detected in each sub-technique (i.e., row sum). Hence, diagonal entries show the method’s sensitivity.

**Figure 8 sensors-21-02705-f008:**
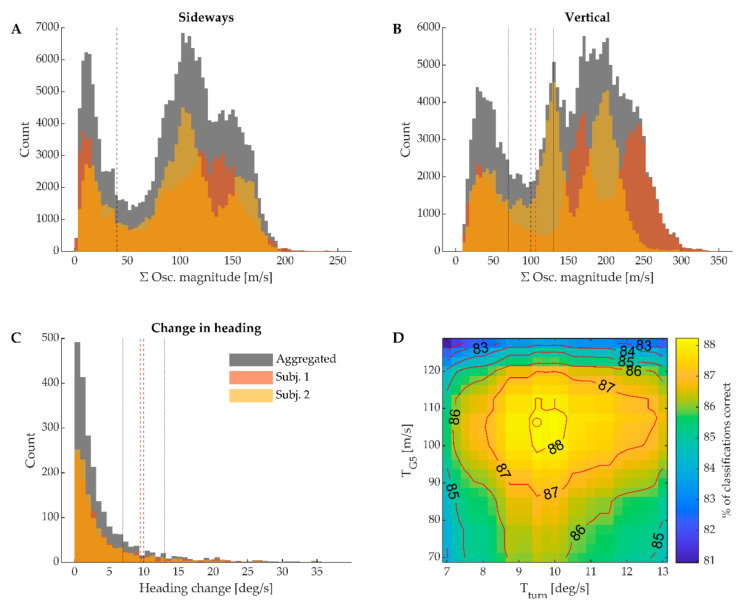
Top row: Histograms showing the distributions the summed frequency magnitudes of the head’s velocity in the sideways (panel (**A**), frequencies between 0.5–1.0 Hz) and vertical (panel (**B**), frequencies between 0.5–1.5 Hz) directions. (**C**) Histogram showing the distribution of change in skiing direction per technique cycle. In (**A**–**C**), yellow and red bars show the individual participants’ contributions to the aggregated distribution (gray bars), dashed black lines show the thresholds used in the hard decision rules classification, dotted lines ((**B**,**C**) only) show the range that was explored in the numerical optimization, and the dashed red line shows the value maximizing the number of correct classifications. (**D**) Result of the numerical optimization of different thresholds (*T*_G5_ and *T*_turn_). The contour plot shows how the number of correctly classified cycles (sum of diagonal entries on the confusion matrix, Figure 6A) changed when *T*_G5_ and *T*_turn_ varied. The number of correct classifications is presented as a percentage of the total number of detected cycles in the video-based validation. Changes in the threshold used in panel A could not be included in the same analysis because this would have changed the number of cycles detected with the automated algorithm, which would require a manual re-assessment of falsely detected and/or not detected cycles for each threshold tested.

## Data Availability

The data presented in this study are available on request from the corresponding author. The data are not publicly available because public sharing of individual data was not included in the written informed consent given by the participants.

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
