# Peer review of "Classification of Cross-Country Ski Skating Sub-Technique Can Be Automated Using Carrier-Phase Differential GNSS Measurements of the Head’s Position"

_sensors, 2021, doi:10.3390/s21082705_

Round 1
Reviewer 1 Report
The article presents interesting research and very well prepared. Test procedure is clear and justified, study data are rightly chosen and sufficient, authors correctly described the study. It is important that the test procedure can be reproduced by other researchers. Text is written in a logical and thoughtful way, creating a coherent whole, in accordance with the writing regime of scientific paper (IMRaD). The method of presentation, comprehensive introduction and very interesting and thoughtful practical examples deserve praise. Authors did a very good job - the text reads very well, and I have no comments.
Author Response
Thank you for taking the time to review our manuscript. We have prepared a rebuttal letter (see attached pdf) where we explain the main changes to the revised manuscript, and give point-by-point responses to your (and the outher reviewers') comments.

Reviewer 2 Report
The aim of this paper was to determine the classification accuracy of a method based on a carrier-phase differential global navigation satellite system for automated determination of sub-technique and cycle characteristics in cross-country ski-ing.
It is highlighted that stationary phases are often used for IMU drift correction. The amount of drift is also depended on the specifications of the IMU and calibration. It might be good to reflect on the growing field of research that aims to develop a drift-free IMU.
Estimation of “positional” information for the treadmill was done by numerical integration of the treadmill’s instantaneous velocity. It would be good to state what the accuracy (error) is of the velocity given by the treadmill, so that error propagation can be estimated.
Please provide information about the selected window size and overlap for the STFT. It is unclear what the argumentation is for setting these values.
State the rational of using a feed-forward neural network classifier (NNC).
Please also state the rational of using a threshold method with a NNC method. How is this threshold set and how might this threshold hold across different conditions and skiers? Some good results are shown regarding the threshold in the results section, but clearer introduction will help the readability of the paper.
The discussion highlights the key issues that need to be discussed. Some further suggestions regarding the discussion are made:
- Please reflect on the use of threshold methods at an initial stage of classification.
- It will be good to also further reflect how different placements of sensors (in general) might change the outcomes.
The discussion can also benefit from expanding the comparison to other methods that have been used, which also applied head-mounted sensors. Most of these will be on gait, but it is good to highlight how this work sits in the broader field. It is especially useful to compare the results with those papers that aim to have infrastructureless tracking (no GPS), as it showcase to the reader that these approaches (e.g. Pedestrian Dead Reckoning) are still very limited in terms of validation for complex tasks, such as as skiing. It highlights the utility of GNSS for more detailed analysis.
A key concern is that the sample groups produces very repetitive patterns, due to the level of experience. This would indicate that the accuracies obtained might also be found with other methods. More importantly it could indicate that the dataset itself might be rather homogenous. Further description of datasets can be useful to determine some of generalisability of the data. Alternatively, comparisons with other techniques provides a more robust determination of the need for NNC to analyse this dataset. The “Tuck” detection works great with a simple threshold. The authors need to state why certain techniques are used and how they compare with other approaches.
Author Response

(The authors gave the same response as above.)

Reviewer 3 Report
Dear Authors,
Thank you very much for your interesting, well written manuscript. I will be able recommend publication in Sensors once you have addressed the following issues:
Major issue:
- Please discuss whether or not differences are to be expected when applying your method developed and validated for roller skiing to “normal”, i.e., snow-based skiing. Personally, I do not expect to much of a difference here due to the similarly of GNSS signal characteristics in both settings, as opposed to IMU data. However, this needs to be addressed by you in your manuscript.
Minor issues:
- Lines 108/109: Please add whether the downsampling was done by averaging or by decimating of adjacent values. That choice will have an impact on noise level. Moreover, please motivate why you have chosen a SD +-30 mm for dGNSS noise simulation, e.g. provide an adequate reference. In [12], that explicit value does not seem to be given.
- Line 182: Overall extra mass of 940 g is much for a trained athlete. Please discuss whether this could have affected the head (or whole body) movement, thus also affecting movement characteristics in different skiing techniques G2–G5.
- Line 424: Please add also GLONASS.
Best regards
Your reviewer
Author Response

(The authors gave the same response as above.)

Reviewer 4 Report
Dear Authors,
the topic of the presented manuscript is relevant and potentially interesting to a broader audience. The methodology is scientifically sound and clearly presented. The conclusions are supported with the provided results and the limitations of the methodology are properly discussed.
Minor concerns, mostly related to manuscript formatting, are as follows:
The submitted manuscript needs reformatting, in order to follow the target journal demands:
- The abstract is in an awkward form. Please consider editing. In particular, the journal explicitly states that you should omit headings in the abstract (i.e., Purpose, Methods…).
- The manuscript body text should be justified.
- Empty lines like between Lines 103 and 104 should be omitted. If not the first paragraph of a section, new paragraph should have a right indentation.
- Consistent font size for body text (text from Line 82 forward seems to be smaller the text in Lines 79 and before).
- Differential GNSS (dGNSS) should be used for the first occurrence of dGNSS in text.
- (a) in Figure 5 should be removed.
Does classic skiing indeed have stationary phases, like argued in Lines 46-47 and referred to reference [5], which is, by the way, missing in the reference list?
The number of participants included in the indoor part of the study must be presented.
Techniques named G2, G3, and G3 should be explained, along with the presented conditions (Lines 101-103).
Minor noticed grammar errors:
‘The’ in ‘the different techniques’ in Line 71 should be omitted.
Author Response

(The authors gave the same response as above.)

Reviewer 5 Report
The article presents methods to classify ski skating sub-techniques and determination of cycle length and frequency, which are based on the signal from dGNSS.
The motivation of using GNSS instead of wearable sensor is reasonable. The proposed methods were validated in an outdoor skiing test case, which showed high classification results in three main gears (G2-G4) athough the number of persons who participated in the outdoor test is small, i.e., 2 persons. However, the result shows a potential to be applicable to similar skill levels.
For broad readers, it might be better to describe ski skating and classical skiing. Actually, cross country skiing is not popular in my country, and thus I did not have the knowledge unless I searched in the Internet.
Additionally, I would like to hear the reason why neural network-based classifier was chosen from various classifier models such as support vector machines, random forest classifier, etc.
Author Response

(The authors gave the same response as above.)

Round 2
Reviewer 2 Report
The authors have addressed the comments appropriately. There is only one minor comment remaining. I appreciate that the authors do not want to add a paragraph on conventional gaits to the discussion, but it would be good to state where the overall field of infrastructureless vs infrastructure tracking sits. One or two review state-of-the art papers might be suitable enough, as this will help position the sport-specific work in the broader field.
Author Response
We appreciate your suggestion to put our work in a broader context by referencing methods commonly applied to conventional gaits, and to point at inherent limitations in step length calculations with some of these methods. We have revised our manuscript to accommodate this suggestion. First, we have added one sentence to the introduction (lines 75-77) along with a new reference (Terrier and Schutz, 2005). This study assessed the possibility of using dGNSS to measure step length in walking. Second, we have revised the discussion (line 489-504) by adding a paragraph discussing methodological issues of cycle length calculations in both cross-country skiing (citing IMU-based methods used in classical skiing) and walking (citing the review by Diez et al. (2018) on IMU-based step-length calculations in conventional gaits). Specifically, the paragraph reads:
“Although IMU-based approaches to determine cycle length has been successfully applied to some classical style skiing techniques that include stationary phases [5], this approach would not be transferable to ski skating techniques. In fact, obtaining cycle length from IMU measurements alone is non-trivial even for conventional gaits with stationary phases [28]. Accurate position sensing technologies such as dGNSS has an intrinsic advantage over IMU-based calculations of cycle length, because displacement can be measured directly. This advantage is not limited to sliding gaits, and has also been suggested for conventional gaits [15].“
We hope you consider our revisions appropriate.